# CaRE: Continual Real-time Unlearning with Ensured Preservation of LLM Knowledge

## Abstract

As concerns grow over the issue of large language models (LLMs) inadvertently internalizing sensitive or erroneous information, unlearning—the selective removal of undesired knowledge—has been drawing an increasing amount of attention. Existing approaches to unlearning fail to account for scenarios requiring immediate processing of knowledge removal requests, leaving services that rely on LLMs vulnerable to prolonged leakage of sensitive information while the process of unlearning is underway. Moreover, when such requests occur not just once, but continuously over the period of deployment, current methods cause LLMs to suffer increasingly degraded utility performance with the processing of each request. To address these issues, we propose **C**ontinu**a**l **R**eal-time Unlearning with **E**nsured Preservation of LLM Knowledge (**CaRE**). Prior to LLM deployment, we train an unlearning sentence embedder with a synthetically generated dataset designed to enable the formation of sharp decision boundaries for determining whether a given input query corresponds to any forget requests in the database. At inference, an embedding is generated for the input query and compared with the embedding of each forget request using a distance metric and the maximum score is compared to a threshold which is used to decide whether to answer the query or to refuse. Since our method does not modify any weights of the language model, it avoids catastrophic forgetting and is able to achieve near perfect knowledge preservation after an arbitrary number of updates. Our experiments on four benchmarks demonstrate that **CaRE** achieves a superior balance of forgetting and knowledge preservation over all existing methods in the continual setting while also being the only method capable of processing forget requests in real-time.

## 1 Introduction

With the rapid advancement of Large Language Models (LLMs), their applications have been swiftly expanding across society and into various aspects of daily life. However, many unforeseen challenges regarding their reliability are also coming to light (Ji et al., 2023; Chang et al., 2024; Gallegos et al., 2024; Zhao et al., 2024). One of these issues is the need for a way to reliably erase targeted pieces of information from an LLM in a localized manner. During the pre-training or finetuning of LLMs on large-scale datasets, they are at risk of incorporating and disseminating sensitive information (Carlini et al., 2021), including copyrighted or privacy-related content (Das et al., 2025), as well as incorrect knowledge (De Angelis et al., 2023). Not only does this raise significant ethical issues, but it also entails legal risk for developers of LLMs as 'the Right to be Forgotten' is mandated by regulations such as the General Data Protection Regulation (GDPR) (Mantelero, 2013) in the EU and the California Consumer Privacy Act (CCPA) (de la Torre, 2018) in the US.

As a countermeasure, LLM unlearning (Jang et al., 2022; Liu et al., 2025) has been introduced with the aim of efficiently removing inappropriate information while preserving the existing knowledge and capabilities of the model—without requiring full retraining. Most approaches to LLM unlearning (Zhang et al., 2024; Jia et al., 2024) apply a training algorithm to the LLM, utilizing a *forget set*—the data to be removed—and a *retain set*—the data used to preserve the model's utility. The training objective is typically to either maximize the loss on the original input–output pairs of the *forget set* or minimize the loss on the same inputs paired with refusal responses, while also minimizing the loss on the *retain set* to preserve the model's knowledge with respect to items that are not the subject of any forget requests.

However, existing approaches remain inadequate for deployment in real world scenarios for the following reasons. First, prior methods have mainly been designed and evaluated under the assumption of a single, unchanging set of forget requests, and their performance is unlikely to hold up under circumstances that require the sequential processing of a continual stream of new requests. In particular, most existing methods operate by modifying the weights of the target LLM, which impairs its general ability and knowledge due to the phenomenon of catastrophic forgetting (Luo et al., 2023), and this problem is only compounded by the accumulating forget requests in the continual setting. Furthermore, while some prior studies have explored the continual setting, they still suffer from catastrophic forgetting on non-target data, and exhibit poor generalization with respect to paraphrased variants of questions in the *forget set*. This indicates that with existing continual unlearning methods, LLMs not only experience utility degradation, but also fail to fully eliminate the information specified in the forget requests. Finally, the nature of many such requests necessitates immediate action to prevent further harm—as in the case of sensitive or dangerous information—while most existing approaches rely on expensive and time-consuming optimization procedures applied to the target LLM. In addition to the heavy cost in time and compute incurred by the training process itself, optimization-based methods typically entail some degree of hyper-parameter search (Bergstra & Bengio, 2012) to find an acceptable balance between effectiveness of forgetting and preservation of knowledge. In cases where a *retain set* is required, securing a sufficient quantity of high-quality data fit for the task can cause even further delay (Gao et al., 2025).

To tackle these challenges, we introduce **C**ontinu**a**l **R**eal-time Unlearning with **E**nsured Preservation of LLM Knowledge (**CaRE**). Prior to LLM deployment, **CaRE** trains an unlearning sentence embedder on a synthetically generated dataset with hard negatives designed to enable fine-grained classification between user queries related to the *forget set*, and those that are unrelated. After the LLM is deployed, **CaRE** continuously adds the embeddings of any received forget requests to its embedding database in real-time and compares them with the embedding of the current user query. Then based on this comparison, we decide whether the LLM should provide a response to the user query or refuse. Importantly, since the unlearning embedder does not require any additional training post-deployment—and in particular, does not need to use either the *forget set* or the *retain set* for training, the entire process achieves significantly faster unlearning compared to prior approaches. Moreover, because the weights of the LLM remain unmodified, **CaRE** allows for near perfect utility preservation. As a result, not only does our method substantially outperform all other unlearning methods in the continual setting (which is the setting most relevant to real world applications), it is the first method we are aware of that is capable of processing ongoing forget requests in real-time with minimal degradation of model performance as requests accumulate over time.

In summary, the contributions of our work are as follows:

- We introduce **CaRE**, an unlearning framework that entails virtually no overhead for processing new forget requests and thus constitutes the first unlearning method capable of handling continual, sequential forget requests in real-time.

- Through experiments across multiple benchmark datasets, we demonstrate that by leaving the weights of the LLM unmodified, **CaRE** is able to largely circumvent the catastrophic forgetting problem faced by existing methods and achieve near perfect preservation of LLM knowledge, even after processing a long succession of continual forget requests.

- We demonstrate superiority over prior state-of-the-art (SOTA) unlearning methods in additional aspects such as the ability of our method to generalize to any unlearning task after training on a single dataset (whereas existing methods typically require retraining on every new *forget* and *retain set*), and robustness to paraphrased variants of sentences in the *forget set*.

## 2 RELATED WORK

**Conventional Unlearning.** Methods that only use the *forget set* for training are called Gradient Ascent (GA) (Jang et al., 2022). These methods train the target LLM to minimize a loss on the *forget set* defined as the positive log likelihood of the text in the *forget set*, thereby minimizing the likelihood of generating the information contained in the *forget set*. Other methods add to this loss by including a term for the negative log likelihood of the text in the *retain set*, which acts as a regularizer forcing the LLM to not only forget the information in the *forget set* but to also explicitly remember the information in the *retain set*. These methods are known as Gradient Difference (GradDiff) (Liu

et al., 2022). A third approach, called Preference Optimization (PO) (Maini et al., 2024), uses a loss that encompasses terms for both the *forget set* and the *retain set*, but instead of using the positive log likelihood on the *forget set*, it uses the negative log likelihood on alternate refusal responses to the questions in the *forget set*. Negative Preference Optimization (NPO) (Zhang et al., 2024) uses the loss from Direct Preference Optimization (DPO) (Rafailov et al., 2023) but with only negative examples (instead of pairs of positive and negative examples). More recent work includes SOUL (Jia et al., 2024), which is not of itself a distinct unlearning method, but rather an improvement that adds second-order optimization to existing methods. These methods tend to have weak performance on knowledge preservation metrics as modifying weights inevitably results in catastrophic forgetting.

**Weight Preserving Unlearning.** Existing approaches that avoid modifying LLM weights include In-Context Unlearning (ICUL) (Pawelczyk et al., 2023) which adds data points from the *forget set* with perturbed labels as in-context examples to the LLM prompt, and guardrail methods (Thaker et al., 2024) that add a filtering step by querying an auxiliary LLM to detect whether the output of the target LLM is related to any data in the *forget set*. These methods generally have low performance except for very large foundation models and they are not scalable as the increasing size of the *forget set* will eventually cause issues due to context length limitations (Liu et al., 2023). Perhaps the method that bears the greatest resemblance to our own is GUARD (Deng et al., 2025). This method also trains a model to classify user queries as being either related or unrelated to the *forget set*. However, the classifier they use is specific to the *forget set* it was trained on and thus needs to be retrained for every new set of forget requests, which precludes the possibility of real-time unlearning and makes it less suitable for the continual setting.

**Continual Unlearning.** Two methods that are particularly relevant to the present work are O3 (Gao et al., 2025) and UniErase (Yu et al., 2025), both of which were designed specifically to address unlearning in the continual setting. The former works by training an orthogonal low-rank adapter (LoRA) (Hu et al., 2021) to unlearn the information in the *forget set*, and then trains an out-of-distribution (OOD) detector to determine how much weight to give to the adapter during inference based on how close the input query is to the data in the *forget set*. The latter method adds an unlearning token "<UNL>" to the tokenizer vocabulary of the LLM and uses prompt tuning (Lester et al., 2021) to train the model to output refusal responses whenever an input query is followed by "<UNL>". It then uses model editing methods (Meng et al., 2022) to modify the weights of the LLM such that when questions from the *forget set* are input to the language model, it generates "<UNL>" as the first token. As these methods both modify the weights of the target LLM (or its adapter), they are still subject to the problem of catastrophic forgetting.

# 3 METHOD FOR REAL-TIME CONTINUAL UNLEARNING

To guarantee not only the preservation of LLMs' existing capabilities, but also to enable effective real-time processing of successive forget requests in a continual unlearning setting, we propose **CaRE** (**C**ontinual **R**eal-time Unlearning with **E**nsured Preservation of LLM Knowledge). Our method begins by training an unlearning sentence embedder $U$ that learns to generate embedding vectors for user queries q and forget requests $f$ whose distance can be used to form a decision boundary for whether to answer the query or to refuse (Sec. 3.2). After the deployment of the LLM, we perform real-time unlearning by asynchronously updating the *forget set* (via generating embeddings for new forget requests $U(f)$ as they are received and adding them to our embedding database) and handling user queries to the LLM through our proposed inference pipeline in conjunction with the trained $U$ (Sec. 3.3). The overall framework of **CaRE** is illustrated in Figure 1.

## 3.1 PROBLEM FORMULATION

To formalize our task, we begin by denoting $D$ as the entirety of the data used to train the large language model $G$ that serves as the starting point for unlearning. $D$ can be partitioned into two splits, the forget split $D_f$ and the retain split $D_r$, where the former represents all the data that needs to be forgotten and $D_r = D \backslash D_f$ represents the rest of the data, which needs to be preserved by the language model. The gold standard of what we are trying to achieve with unlearning is a model $G^*$

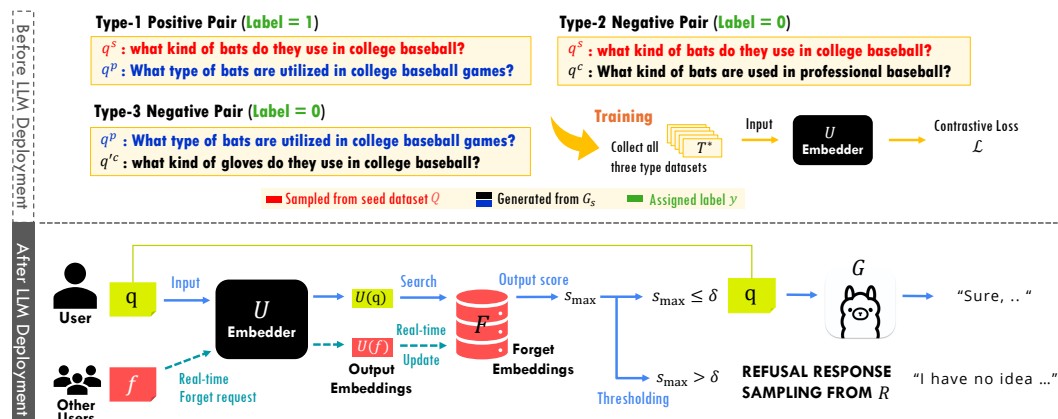

Figure 1: An overview of the **CaRE** framework. **CaRE** consists of a training phase carried out prior to deployment (upper part) and a three-step inference process after deployment (lower part). In the training phase, the embedder $U$ is trained on three types of synthetic data generated from a seed dataset (training does not require any data from the *forget set* or *retain set*). For inference, real-time continual unlearning is enabled through three steps: (**i**) embed q, embed-and-store $f$, (**ii**) retrieval and thresholding, and (**iii**) decision on whether the LLM responds or refuses. Since the LLM's weights remain unchanged, we are able to maintain a high level of utility preservation.

that has been trained in the same manner as $G$, but on $D_r$ only. Such a model would not contain any knowledge of the data in $D_f$ since it was never trained on $D_f$ and it could be expected to contain roughly the same amount of knowledge about $D_r$ as $G$, since it is assumed to have undergone the same training process on those data points.

In most real world applications, $G^*$ is just a theoretical ideal that cannot be obtained in practice since modern LLMs are too large and costly to retrain from scratch. Hence, this objective is approximated by performance metrics on $D_f$ that gauge how effectively the data in $D_f$ has been forgotten and performance metrics on $D_r$ that measure how well the rest of the data has been preserved. Most unlearning techniques involve modifying the weights of $G$ to obtain an approximation $\widehat{G} \approx G^*$, which subjects the language model to heavy drops in performance on $D_r$ as the weight updates give rise to catastrophic forgetting (Luo et al., 2023), a problem that is worsened in the continual setting described below. Our method on the other hand, does not modify $G$ at all, thus preserving its existing knowledge in tact and leaving the potential for achieving the same performance on $D_r$ as $G$ an open possibility.

**Continual setting.** To closer align our task with scenarios likely to be encountered in the real world, we additionally extend the unlearning task to the continual setting where the forget requests arrive successively and need to be processed cumulatively in sequence. Hence, we start with an initial partition $D_{f_0}$, $D_{r_0} = D \backslash D_{f_0}$ to which we apply our unlearning techniques and evaluate. Then the *forget set* is expanded to include new requests resulting in a new partition $D_{f_1}$, $D_{r_1} = D \backslash D_{f_1}$ such that $D_{f_0} \subset D_{f_1}$ and we perform further unlearning on the same model to reflect the additional requests and evaluate once more. The goal is to maintain high performance on the forget and retain objectives over each stage until the final set of forget requests and final partition $D_{f_N}$, $D_{r_N} = D \backslash D_{f_N}$. If finetuning is applied to $G$ post-deployment to add new information, $D$ itself may also expand, but for simplicity we assume that $D$ is fixed.

Most existing unlearning methods use the entire forget split for training, hence the *forget set* used for training is simply $D_f$. Methods that also make use of the retain split for training cannot use the entire split since it is too vast, so they typically use a small subset consisting of counterexamples to the *forget set* which is termed the *retain set* $D_{retain} \subset D_r$. For evaluation, again typically the entire forget split $D_f$ is used to test forgetting effectiveness, whereas to test preservation of knowledge, various subsets of $D_r$ are used, including the *retain set* as well as utility datasets that are completely unrelated to the *forget set* to test general knowledge capacity, such as "World Facts" in the TOFU benchmark (Maini et al., 2024) and WinoGrande (Sakaguchi et al., 2019) in the RETURN benchmark (Liu et al., 2024).

## 3.2 PRE-DEPLOYMENT TRAINING

We now describe the first step of the **CaRE** framework, which involves training the unlearning sentence embedder $U$. Before deployment of the large language model $G$, it is unknown what removal requests $f$ may arise, or what queries q may be issued to $G$. Therefore, $U$ must learn a representation that effectively distinguishes and generalizes over any possible future q and $f$, taking this uncertainty into account. It must also be robust to variations of the *forget set*, e.g., paraphrased sentences that convey the same information as those in the *forget set* should still trigger refusal to respond. To meet the above requirements, we build training data of three types through the following process.

First, we collect the questions from a seed QA dataset $Q = \{q_1^s, q_2^s, \ldots, q_n^s\}$, e.g., Natural Questions (Kwiatkowski et al., 2019), where each $q_i^s$ represents the question from the question-answer pair $(q_i^s, a_i^s)$. For each question $q^s$, we apply transformations as illustrated in Figure 1 to generate two variants of the question, $G_s(\tau_1(q^s)) = (q^p, q^c)$, where $\tau_1(\cdot)$ is an input prompt template for a surrogate LLM $G_s$. Here $q^p$ represents a paraphrased variant of $q^s$ and $q^c$ represents a contrastive variant. $q^p$ is thus a rephrasing of $q^s$ that should elicit the same response from the target LLM $G$. Coupled with $q^s$, $(q^s, q^p)$ constitutes a positive pair with label $y^p = 1$, which we term type-1 data. In contrast, $q^c$ is a question designed to exhibit high lexical or syntactic overlap with $q^s$ but differ in semantic meaning. Together with $q^s$, the pair $(q^s, q^c)$ serves as a hard-negative example with label $y^c = 0$, which we term type-2 data. Following the same procedure, we obtain the contrastive sample of $q^p$ via $\tau_2(\cdot)$, denoted as $q'^c = G_s(\tau_2(q^p))$, which paired with $q^p$ as $(q^p, q'^c)$ forms an instance of type-3 data labeled with $y'^c = 0$, thereby functioning as an additional hard-negative sample along with the type-2 data. We apply the three types of data augmentation to every sample in $Q$, and construct the dataset $T^* = \{[(q_i^s, q_i^p), y_i^p], [(q_i^s, q_i^c), y_i^c], [(q_i^p, q_i'^c), y_i'^c]\}_{i=1}^n$ for training the embedder $U$. We use $T^*$ to finetune a pre-trained sentence embedding model (Reimers & Gurevych, 2019) using the following contrastive loss (Hadsell et al., 2006):

$$\mathcal{L}(T) = \frac{1}{2|T|} \sum_{(q,q',y) \in T} \left[ y \cdot d_U(q, q')^2 + (1-y) \cdot \max\left(0, m - d_U(q, q')\right)^2 \right], \tag{1}$$

where $d_U$ denotes a distance metric in the embedding space of $U(\cdot)$, which is the cosine distance in our case defined as $d_U(q, q') = 1 - \frac{U(q) \cdot U(q')}{\|U(q)\| \|U(q')\|}$. $T \subset T^*$ is a batch of samples from the training dataset and $m$ is an appropriately chosen margin. The loss serves to decrease the distance between positive examples and increase the distance between negative examples up to the margin $m$. The hard-negative samples in our dataset are designed to represent difficult edge cases, thereby enabling the embedder to form more fine-grained and precise decision boundaries in the embedding space. It should be noted that all of the above training is conducted without requiring either the *forget set* or *retain set*, and that it is carried out prior to the deployment of $G$. After deployment, the single trained $U$ model can operate across any given forgetting task and domain without any additional training and its effectiveness is not limited to any particular *forget* and *retain set*.

## 3.3 POST-DEPLOYMENT INFERENCE

Once $G$ is deployed, **CaRE** performs unlearning and inference through the following three steps. (**i**): Given the $m$-th forget sample $f_m$, its embedding $f_m^{\mathrm{emb}} = U(f_m)$ is generated and stored in the set of forget embeddings $F$. The update of $F$ is carried out immediately in real-time upon arrival of $f_m$ and can be expressed as

$$F = \{f_1^{\mathrm{emb}}, f_2^{\mathrm{emb}}, \ldots, f_{m-1}^{\mathrm{emb}}\} \quad \Rightarrow \quad F \leftarrow F \cup \{f_m^{\mathrm{emb}}\}. \tag{2}$$

This instantaneous operation constitutes the entirety of our unlearning process post-deployment and stands in stark contrast to the heavy optimization procedures employed by other methods to unlearn a given set of forget requests. Asynchronously, whenever a user query q is input to $G$, it is projected into the embedding space as $q^{\mathrm{emb}} = U(q)$. (**ii**): For each embedding $f_i^{\mathrm{emb}}$ in $F$, we compute its cosine similarity score $s_i$ with respect to $q^{\mathrm{emb}}$, and obtain the score set $S = \{s_i\}_{i=1}^m$, where $s_i \in [-1, 1]$. Using $S$, we identify the element $f_j \in F$ most related to q by taking an element with the maximum score $s_j = s_{\max}$, and check whether it exceeds a given threshold $\delta$. In this process,

the user queries sent to the LLM and requests for information removal are all handled continuously and in real-time, without mutual interference. (**iii**): The final response $r_{\text{res}}$ returned to the user is defined as follows:

$$r_{\text{res}} = \begin{cases} G(\text{q}), & \text{if } s_{\max} < \delta \,, \\ \text{a sampled element from } R, & \text{if } s_{\max} \geq \delta \,, \end{cases} \tag{3}$$

where $R$ is a predefined set of refusal expressions such as "I don't know" or "I can't answer that question". If $s_{\max} < \delta$, we determine that q is unrelated to any information in the current *forget set*, and thus return the regular generated output for q using $G$. In contrast, if $s_{\max} \geq \delta$, we determine that q is closely related to some information in the *forget set* and therefore decline to answer q. In this case, a refusal response is sampled from $R$ and returned as $r_{\text{res}}$ (Appendix F). Note that the parameters of $G$ are not modified at any step of this process. This guarantees knowledge preservation within $G$ thereby preventing the occurrence of catastrophic forgetting, which is key to our method being able to maintain such high performance on the retain and utility datasets after processing an arbitrary number of successive forget requests.

## 4 EXPERIMENTS

### 4.1 EXPERIMENTAL SETUP

**Benchmarks.** We conduct unlearning experiments in the continual setting using four widely used benchmarks. **(1)** *Privacy Data Unlearning*: The RETURN benchmark (Liu et al., 2024) consists of synthetically generated question-answer pairs related to real world individuals with Wikipedia pages. The goal is to forget selected details (not all) about a subset of the individuals. We posit a scenario where out of the 30 target individuals, three individuals issue forget requests at each stage, resulting in a total of 10 stages of continual unlearning. **(2)** *General Science Knowledge Unlearning*: We adopt the setting in Gao et al. (2025) which uses a subset of the ScienceQA dataset (Lu et al., 2022) as the *forget set* to sequentially unlearn four scientific topics: biology, physics, chemistry, and economics. At each stage, one topic is added to the *forget set* and the remaining topics make up the *retain set*. **(3)** *Fictitious Authors Unlearning*: TOFU (Maini et al., 2024) is an unlearning benchmark that fine-tunes a pre-trained language model on QA pairs about completely fabricated authors to ensure that none of the data in the *forget set* exists in the pre-training data. The task is then to unlearn information about a selection of the fake authors. We divide the authors into three groups, resulting in a three-stage continual unlearning setup. **(4)** *False Information Unlearning*: TruthfulQA (Lin et al., 2021) is a benchmark designed to assess whether LLMs provide factually grounded answers to misleading questions across diverse topics (i.e., whether they avoid generating misinformation). We adopt a continual unlearning setting in which all the questions are partitioned into three stages and used as the *forget set*. Further details about the evaluation datasets can be found in Appendix C.1

It should be noted that for the *forget set* used for evaluation, we replace the questions with paraphrased variants as this is a more realistic assumption for real world use cases and using the same questions verbatim from the original *forget set* would be trivial for our method to solve with 100% accuracy by setting the decision boundary threshold $\delta$ to 1. Also, for each benchmark we add a synthetically generated *near utility* dataset containing examples designed to be similar in appearance to sentences in the *forget set*, but distinct in meaning (and hence should not be subject to removal—they are edge cases designed to test the locality of the forgetting mechanism). The detailed procedure for generating these datasets is outlined in Appendix E.

**Evaluation Metrics.** As our method does not modify any weights of the LLM, it does not alter the probability distribution output by the LLM, which renders probability-based metrics such as the Truth Ratio (Maini et al., 2024) meaningless for our case. Hence for most evaluation datasets we use ROUGE-L (Lin, 2004) to measure the similarity between the generated response and the ground truth answer. In cases where we are able to extract an exact answer from the generated response using simple parsing, such as the WinoGrande dataset (Sakaguchi et al., 2019) and the ScienceQA benchmark (Lu et al., 2022), we calculate accuracy using an exact match criterion.

**Baselines.** We selected GA (Jang et al., 2022), GradDiff (Liu et al., 2022), PO (Maini et al., 2024), NPO (Zhang et al., 2024), SO-PO (Jia et al., 2024), GUARD (Deng et al., 2025), O3 (Gao et al.,

2025), and UniErase (Yu et al., 2025) as our baselines. Base indicates the target model prior to unlearning, which serves as an upper bound for knowledge preservation performance. UniErase only works on data given in (subject, relation, object) triplet form i.e. questions and answers about people, so we exclude it from our experiments for TruthfulQA and ScienceQA.

The training configuration of $U$, the details of $\tau_1$ and $\tau_2$, and results for other models are presented in Appendices C.2, D, and H respectively.

## 4.2 PRIVACY DATA UNLEARNING

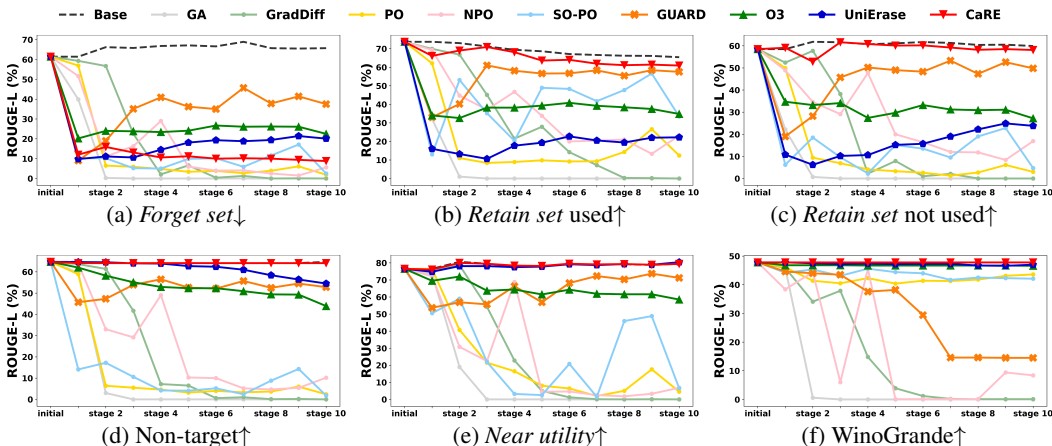

Figure 2: Continual unlearning results on RETURN. (a) indicates performance on the unlearning target, while (b)–(f) indicate performance on data that we aim to preserve (details in Appendix C.1).

Figure 2 presents our experimental results on the RETURN benchmark. The gradient-based and preference optimization methods exhibit a strong tendency towards overforgetting—they are successful in removing the knowledge related to the *forget set* but at the cost of significant degradation in performance on unrelated knowledge. We can clearly see a sharp drop-off from the base model as the stages progress—as expected due to catastrophic forgetting. GUARD, O3 and UniErase preserve knowledge to some extent, but fail to sufficiently remove the target knowledge. **CaRE**, on the other hand, achieves effective removal of the data from the *forget set* with negligible degradation in performance on the other datasets across all ten stages of evaluation.

## 4.3 GENERAL SCIENCE KNOWLEDGE UNLEARNING

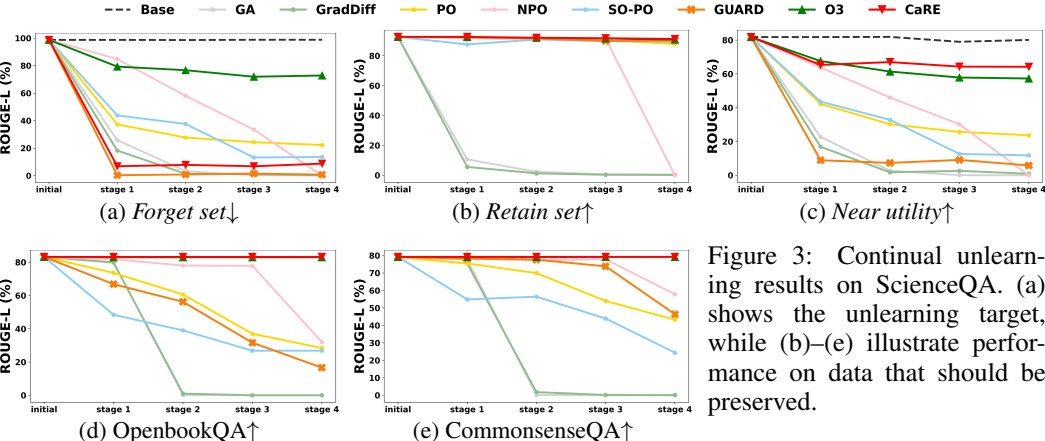

(a) *Forget set↓*

(b) *Retain set↑*

(c) *Near utility↑*

(d) OpenbookQA↑

(e) CommonsenseQA↑

Figure 3: Continual unlearning results on ScienceQA. (a) shows the unlearning target, while (b)–(e) illustrate performance on data that should be preserved.

Figure 3 presents our results on the ScienceQA benchmark. The only method that is able to maintain comparable performance to **CaRE** on the knowledge preservation datasets across all stages of

evaluation is O3. However, we can see that its performance on the *forget set* is unusually poor. We found that this is due to O3 being unable to generalize to paraphrased variants of the questions in the *forget set*. While it is able to achieve much lower scores of 20.7%, 4.6%, 10.1%, and 11.8% across the four stages of the original *forget set*, it is surprisingly brittle against even slight changes in wording and thus cannot be said to have truly forgotten the information in the *forget set*. So again **CaRE** is the only method able to achieve effective forgetting while maintaining near perfect knowledge preservation across each stage of evaluation.

## 4.4 FICTITIOUS AUTHORS UNLEARNING

Table 1: Results on the TOFU benchmark. **F.G.** (*forget set*), **R.T.** (*retain set*), **N.U.** (*near utility*), **R.A.** (Real-Authors), and **W.F.** (World Facts) are reported; the best results are highlighted in **blue**, and the second-best are underlined, excluding near-zero values on **F.G.** caused by over-forgetting.

| | TOFU dataset for LLaMA2-7B-chat | | | | | | | | | | | | | | |
| | Stage 1 | | | | | Stage 2 | | | | | Stage 3 | | | | |
| Method | F.G.↓ | R.T.↑ | N.U.↑ | R.A.↑ | W.F.↑ | F.G.↓ | R.T.↑ | N.U.↑ | R.A.↑ | W.F.↑ | F.G.↓ | R.T.↑ | N.U.↑ | R.A.↑ | W.F.↑ |
|---|---|---|---|---|---|---|---|---|---|---|---|---|---|---|---|
| Base | 0.496 | 0.973 | 0.620 | 0.940 | 0.913 | 0.518 | 0.973 | 0.617 | 0.940 | 0.913 | 0.509 | 0.973 | 0.599 | 0.940 | 0.913 |
| GA | 0.390 | 0.715 | 0.574 | 0.855 | 0.821 | 0.211 | 0.320 | 0.488 | 0.576 | 0.785 | 0.003 | 0.003 | 0.005 | 0.000 | 0.006 |
| GradDiff | 0.242 | 0.424 | 0.550 | 0.763 | 0.812 | 0.001 | 0.002 | 0.003 | 0.000 | 0.003 | 0.000 | 0.000 | 0.000 | 0.000 | 0.000 |
| PO | 0.110 | 0.873 | 0.598 | 0.923 | 0.883 | 0.111 | 0.801 | 0.533 | 0.692 | 0.862 | 0.181 | 0.860 | 0.570 | 0.897 | 0.877 |
| NPO | 0.072 | 0.874 | 0.608 | 0.930 | 0.892 | **0.031** | 0.796 | 0.601 | 0.912 | 0.900 | 0.065 | 0.815 | 0.593 | 0.914 | 0.895 |
| SO-PO | 0.094 | 0.837 | 0.586 | 0.899 | 0.896 | 0.118 | 0.808 | 0.592 | 0.922 | 0.868 | 0.120 | 0.791 | 0.562 | 0.916 | 0.873 |
| GUARD | 0.121 | 0.773 | 0.573 | 0.909 | 0.896 | 0.112 | 0.798 | 0.536 | 0.872 | 0.883 | 0.129 | 0.775 | 0.553 | 0.891 | 0.876 |
| O3 | 0.128 | 0.338 | 0.564 | 0.651 | 0.905 | 0.070 | 0.093 | 0.198 | 0.095 | 0.282 | 0.083 | 0.093 | 0.163 | 0.079 | 0.219 |
| UniErase | 0.047 | 0.947 | 0.603 | 0.906 | **0.930** | 0.058 | 0.943 | 0.610 | 0.899 | **0.930** | 0.062 | 0.942 | 0.587 | 0.889 | 0.905 |
| **CaRE** | **0.046** | **0.969** | **0.620** | **0.940** | 0.913 | 0.055 | **0.969** | **0.615** | **0.940** | 0.913 | **0.043** | **0.961** | **0.597** | **0.940** | **0.913** |

Table 1 presents our results on the TOFU benchmark. The only method that appears to remain competitive with our method across all three stages is UniErase. However, the apparent strength of this method—which still lags **CaRE** in overall performance—should be weighed against the inability of UniErase to handle any data that does not conform to its strict (subject, object, relation) format, which is a significant limitation, as well as its inability to process forget requests in real-time.

## 4.5 FALSE INFORMATION UNLEARNING

Table 2 reports the results for TruthfulQA. The objective in this case is to prevent the dissemination of false information contained in the *forget set*. However, minimizing similarity to a particular incorrect answer can be gamed: the model may simply produce a different incorrect response while remaining untruthful. Hence, instead of measuring the similarity of the response to the answers in the *forget set*, we measure its similarity to a

Table 2: Results on TruthfulQA benchmark. **R.F.** (refusal answers), **N.U.** (*near utility*), and **C.Q.** (CommonsenseQA) are reported; best: **blue**; second-best: underlined .

| | TruthfulQA dataset for LLaMA2-7B-chat | | | | | | | | |
| | Stage 1 | | | Stage 2 | | | Stage 3 | | |
| Method | R.F.↑ | N.U.↑ | C.Q.↑ | R.F.↑ | N.U.↑ | C.Q.↑ | R.F.↑ | N.U.↑ | C.Q.↑ |
|---|---|---|---|---|---|---|---|---|---|
| Base | 0.5351 | 0.6919 | 0.8256 | 0.5378 | 0.7067 | 0.8256 | 0.5367 | 0.7006 | 0.8256 |
| PO | 0.9030 | 0.0637 | 0.3790 | 0.9389 | 0.0373 | 0.2968 | 0.9792 | 0.0340 | 0.3243 |
| SO-PO | 0.9019 | 0.2195 | 0.6059 | 0.8634 | 0.3115 | 0.4962 | 0.8216 | 0.3144 | 0.5392 |
| O3 | 0.9869 | 0.3691 | 0.2685 | **0.9980** | 0.2585 | 0.2010 | **0.9995** | 0.3702 | 0.2647 |
| **CaRE** | **0.9942** | **0.6068** | **0.8231** | 0.9882 | **0.6072** | **0.8190** | 0.9855 | **0.5932** | **0.8149** |

set of refusal responses (the pairwise maximum from the set) such as "I don't know" as our indication of success. This inherently restricts our evaluation to methods that are capable of optimizing towards a desired response (i.e. it excludes gradient ascent methods that only optimize away from an undesirable response). From the table we can see again that **CaRE** has much stronger performance than existing methods and that its advantage grows with each stage of evaluation.

## 4.6 ABLATION STUDY

Table 3 presents a comparison of the classification performance of $U$ in the first and final stages of all benchmarks under various ablations, in order to examine the importance of each component of our method.

**Contribution of the Proposed Dataset.** As we can see from the table, training with our datasets (bottom row) improved the F1 score over the baseline (top row) by **15.05%** in the first stage and

Table 3: Classification performance of $U$ on the four benchmarks (RETURN, ScienceQA, TOFU, and TruthfulQA). In Config, the columns indicate whether the three data types (one positive, two negative) setting is used, whether hard-negative samples are used, the size of the training dataset, and which dataset was used as the seed (NQ denotes Natural Questions, TQ denotes TriviaQA). The top row corresponds to the vanilla sentence embedding model without any finetuning, gray regions correspond to settings with all components of our method being applied, and the best **F1** performance is emphasized in **bold**.

| Config | | | | First Stage | | | Last Stage | | |
|---|---|---|---|---|---|---|---|---|---|
| **All types** | **H.N.** | **Size** | **Seed** | **Precision** | **Recall** | **F1** | **Precision** | **Recall** | **F1** |
| ✗ | ✗ | 0k | ✗ | 0.7026 | 0.8538 | 0.7709 | 0.6939 | 0.8954 | 0.7819 |
| ✗ | ✗ | 12k | NQ | 0.4847 | 0.9994 | 0.6528 | 0.5031 | 0.9988 | 0.6691 |
| ✓ | ✗ | 18k | NQ | 0.5535 | 0.9994 | 0.7124 | 0.5512 | 0.9986 | 0.7104 |
| ✗ | ✓ | 12k | NQ | 0.8094 | 0.9727 | 0.8836 | 0.8171 | 0.9493 | 0.8783 |
| ✓ | ✓ | 12k | NQ | 0.8455 | 0.9524 | **0.8958** | 0.8526 | 0.9321 | **0.8906** |
| ✓ | ✓ | 18k | TQ | 0.8497 | 0.9379 | **0.8916** | 0.8548 | 0.9225 | **0.8874** |
| ✓ | ✓ | 18k | NQ | 0.8114 | 0.9780 | **0.8869** | 0.8246 | 0.9581 | **0.8863** |

**13.35%** in the last stage. This improvement can be attributed to the use of contrastive loss on the three types of augmented data, which enables the formation of sharper decision boundaries on unlearning data and thereby enhances classification performance. Dropping any component of our proposed training data configuration still allows our model $U$ to correctly classify queries that should be refused (forgotten) as indicated by the high recall, but it also leads to over-forgetting as indicated by the precipitous drops in precision. Therefore, all components of our proposed training data configuration are necessary to achieve an effective balance between forgetting and knowledge preservation. A more detailed analysis of these results and comparison of classification performance with GUARD are provided in Appendix B.

## 4.7 UNLEARNING EFFICIENCY

In Table 4 we show the average unlearning time per stage on the RETURN benchmark as well as any extra processing time for inference as an average per query for the final stage of RETURN. From the table we can see that **CaRE** exhibits overwhelmingly faster unlearning time compared to all other baselines and is the only method capable of real-time processing of both forget requests and user queries. Due to the required search and retrieval of related forget requests, **CaRE** does incur additional overhead for inference, but as reported in the table this cost is negligible. GUARD comes relatively close, but is not quite real-time for unlearning, while incurring significant latency for inference due to its heavy use of beam search—a cost that will grow dramatically with the size of the LLM being deployed. It should be noted that these times do not include the additional delay incurred by the baselines due to hyperparameter search.

Table 4: Measured efficiency of unlearning and inference post-LLM deployment on RETURN. Our method highlighted in **bold** (gray region).

| Post-deployment efficiency (s) | | |
|---|---|---|
| **Method** | **Unlearning time** | **Inference overhead** |
| GA | 195.6 | 0 |
| GradDiff | 229.5 | 0 |
| PO | 178.8 | 0 |
| NPO | 249.4 | 0 |
| SO-PO | 209.4 | 0 |
| GUARD | 2.8 | 25.5 |
| O3 | 327.6 | 0.05 |
| UniErase | 323.2 | 0 |
| **CaRE** | **0.04** | **0.01** |

## 5 CONCLUSION

We showed that existing LLM unlearning approaches suffer from catastrophic forgetting and are inadequate for the continual real-time processing required in real world settings. To address this, we proposed **CaRE**, which trains an unlearning sentence embedder on a three-type dataset with hard-negative samples, prior to LLM deployment, without requiring a *forget set* or a *retain set*. At inference time, **CaRE** works in three steps to handle new forget requests and user queries in real-time without modifying the LLM weights. Experiments on four benchmarks demonstrate that **CaRE** maintains performance on utility datasets nearly identical to the pre-unlearning base model while achieving effective generalization in forgetting, establishing it as the most reliable method among all baselines and the first method capable of operating in real-time.

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

APPENDIX

This appendix provides supplementary materials and additional experimental results. It is organized as follows:

## A   DISCUSSION ON THE COST OF RETAIN SETS

Table 5: *Retain set* sizes for methods requiring them in unlearning experiments on four benchmarks.

| *Retain set* | RETURN | ScienceQA | TOFU | TruthfulQA | Total |
|---|---|---|---|---|---|
| Size | 150 | 1827 | 3800 | 817 | 6594 |

The *retain set* is a dataset that, paired with the *forget set*, is used by some unlearning methods to train the target LLM. Its role is to act as a regularizer to preserve existing knowledge during training and as such, it consists of a collection of representative examples of the knowledge or information that should be preserved. For example, GradDiff, NPO, PO, and SO-PO all employ a loss on the *retain set* during optimization for unlearning. In GUARD, a classifier is trained by using samples from the *forget set* as positive examples and samples from the *retain set* as negative examples. However, employing a *retain set* necessitates the securing of data of sufficient quantity and quality (Gao et al., 2025), which can be highly time consuming. This introduces an additional source of latency to the post-deployment unlearning process and thus, avoiding reliance on a *retain set* is crucial in real-time scenarios. Our approach does away with the need for a *retain set* and thus entirely dispenses with the cost of collecting and training the datasets shown in Table 5, thereby enabling unlearning that is both efficient and effective.

## B   DETAILS OF THE ABLATION STUDY

In this section, we provide a more detailed explanation of Table 3 and describe Table 6, which compares our classification performance with that of GUARD.

**Impact of the Three Data Types.** We tested the importance of having the three types of data augmentation by training with only two types. As the resulting dataset contained only two thirds (12k samples) the number of samples in the original dataset, we conducted an additional experiment using only 12k samples from the original dataset (with all three data types) to control for the effect of dataset size. As we can see from the table, using only two data types leads to a slight drop in

F1 score and this drop is not due to the reduction in number of samples as the performance of the 12k control dataset does not show a similar drop (and even slightly improves upon the original 18k dataset).

**Effectiveness of Hard Negatives.** To evaluate the impact of generating hard-negative samples for the type-2 and type-3 data, we constructed an alternate dataset where $q^c$ and $q'^c$ were semantically distinct from $q^s$ and $q^p$, but also had no lexical or structural overlap with the latter. Specifically, $q^c$ and $q'^c$ were randomly sampled from the seed dataset excluding $q^s$. Experimental results show that constructing hard-negative samples with our proposed method improves the F1 score by **24.49%** in the first stage and by **24.76%** in the final stage, compared to the case without hard negatives.

**Generalization across Seed Datasets.** To test the robustness of our method across different seed datasets, we tried switching the seed dataset to TriviaQA (Joshi et al., 2017). From the table we can see that switching the seed dataset does not compromise the classification performance and in fact, using TriviaQA shows slightly improved performance over Natural Questions.

Table 6: Classification performance of $U$ and GUARD on the three benchmarks RETURN, ScienceQA, and TOFU.

| Method | First Stage | | | Last Stage | | |
|---|---|---|---|---|---|---|
| | Precision | Recall | F1 | Precision | Recall | F1 |
| GUARD | 0.2049 | 0.9518 | 0.3372 | 0.2708 | 0.9326 | 0.4198 |
| **CaRE** | 0.8436 | 0.9325 | **0.8858** | 0.8572 | 0.9142 | **0.8848** |

**Performance Comparison with GUARD.** From Table 6 we can see that GUARD has high recall but very low precision, indicating a strong tendency towards overforgetting. Thus the classifier is fairly inaccurate and the reason its performance on ROUGE-L and accuracy metrics do not show as severe a drop is that, upon predicting a positive example, it does not block the response of $G$ entirely as we do, but only the words from the retrieved forget request. This is a safer, albeit slower, method of inference that to some extent offsets the weak performance of the classifier, and it could be combined with our more accurate classifier for even more selective blockage of information.

## C EXPERIMENTAL SETUP DETAILS

### C.1 DATASETS AND SPLIT

In this section we provide more details about the datasets used for evaluation (and for training in the case of baselines that use the *forget set* and *retain set* for training).

**(1)** *Privacy Data Unlearning*: For each individual in the RETURN benchmark (Liu et al., 2024), there are 20 synthetically generated QA pairs. Among the 60 sampled individuals, half are designated as targets and the other half as non-targets. For each target individual, 10 QA pairs are assigned to the *forget set* (assumed to contain sensitive information about the target individual) and the remaining 10 QA pairs are assigned to the *retain set* (assumed not to contain any sensitive information about the target individual). The *retain set* is further split into two subsets with 5 QA pairs apiece: *retain set* used, which is used for training (if required by the unlearning method), and *retain set* not used, which is excluded from training. We create 10 stages of continual unlearning by assigning 3 of the 30 target individuals to each stage. For utility data, we use WinoGrande (Sakaguchi et al., 2019).

**(2)** *General Science Knowledge Unlearning*: The ScienceQA dataset (Lu et al., 2022) consists of 26 topics in total. Of these we unlearn biology, physics, chemistry, and economics sequentially in that order. At each stage, all of the remaining topics (that have not been added to the *forget set*) make up the *retain set*. The utility data are drawn from the validation split of CommonsenseQA (Talmor et al., 2018) and test split of OpenbookQA (Mihaylov et al., 2018).

**(3)** *Fictitious Authors Unlearning*: For TOFU (Maini et al., 2024), we divide the 20 authors from the largest forget split, 'forget10' into groups of 10, 5, and 5, resulting in a three-stage continual unlearning setup. The *retain set* consists of 400 samples from authors outside of the *forget set*, and the utility data used are the Real Authors and World Facts datasets.

**(4)** *False Information Unlearning*: From TruthfulQA (Lin et al., 2021) we split all the questions into three stages for continual unlearning and add them sequentially to the *forget set*. The *retain set* is separately generated using prompts for *near utility* as described in Appendix E, while the general utility evaluation is conducted on the CommonsenseQA validation split.

## C.2 TRAINING CONFIGURATION

| Component | Setting |
| --- | --- |
| Base sentence encoder | `sentence-transformers/multi-qa-mpnet-base-dot-v1` |
| Training objective | Contrastive loss (`sentence_transformers.losses.ContrastiveLoss`) |
| Distance metric | Cosine distance (`SiameseDistanceMetric.COSINE_DISTANCE`) |
| Margin | `0.5` |
| Optimizer LR | `2e-5` |
| Warmup steps | `100` |
| Epochs | `1` |
| Batch size | `16` |
| Dataloader | `shuffle=True` |

Table 7: Complete training configuration for the unlearning sentence embedder $U$.

We employed 'multi-qa-mpnet-base-dot-v1' (Reimers & Gurevych, 2019) as the base model for the unlearning sentence embedder $U$. This model has only around 109 million parameters so our training cost is orders of magnitude smaller than existing gradient-based approaches, which train the target LLM. We used 6,000 seed samples from the Natural Questions dataset (Kwiatkowski et al., 2019) to generate the data for training $U$. The parameter $\delta$ was set to 0.9 for RETURN and ScienceQA, and 0.8 for TOFU and TruthfulQA.

In Table 7 we list all the hyperparameter settings we used to train the unlearning sentence embedder $U$. We trained $U$ with three types of augmented data as described above, using the Natural Questions dataset as the seed. In our approach, model training is conducted prior to LLM deployment.

## C.3 UNLEARNING TARGET BASE MODELS

| Benchmark | Model Size | Unlearning Target |
| --- | --- | --- |
| RETURN | 1B | meta-llama/Llama-3.2-1B-Instruct |
|  | 7B | meta-llama/Llama-2-7b-chat-hf |
| ScienceQA | 1B | laurel1313/llama3.2_base_scienceqa |
|  | 7B | gcyzsl/O3_LLAMA2_ScienceQA |
| TOFU | 1B | open-unlearning/tofu_Llama-3.2-1B-Instruct_full |
|  | 7B | open-unlearning/tofu_Llama-2-7b-chat-hf_full |
| TruthfulQA | 1B | meta-llama/Llama-3.2-1B-Instruct |
|  | 7B | meta-llama/Llama-2-7b-chat-hf |

Table 8: Benchmarks, model sizes, and unlearning targets used in our experiments.

For the unlearning target, we used finetuned versions of Llama2-7B (Touvron et al., 2023) on the TOFU and ScienceQA benchmarks and the pre-trained version on all other benchmarks as detailed in Table 8. Experiments were carried out on two A100 GPUs.

## D    PROMPT FOR THREE TYPE DATASET GENERATION

Figure 4 illustrates the core Python code and input prompt templates used for generating the three-type datasets. To enhance generalization, not only interrogative sentences but also declarative sentences are generated with a certain probability. The blue text in the upper figure represents the core prompt of $\tau_1(\cdot)$, while the blue text in the lower figure represents the core prompt of $\tau_2(\cdot)$.

```python
def generate_with_retry(question: str, max_retries=3, delay=2) -> dict:
    paraphrased_form = random.choices(["question", "declarative"], weights=[0.6, 0.4])[0]
    contrastive_form = random.choices(["question", "declarative"], weights=[0.6, 0.4])[0]
    paraphrased_instruction = (
    "Generate the paraphrased sentence as a question that expects an answer."
    if paraphrased_form == "question"
    else "Generate the paraphrased sentence as a declarative sentence (not a question) that
still implies an answer."
    )
    contrastive_instruction = (
        "Generate the contrastive sentence as a question that expects an answer."
        if contrastive_form == "question"
        else "Generate the contrastive sentence as a declarative sentence (not a question)
that still implies an answer."
    )

    system_msg = "You are a helpful assistant that generates sentence variations."
    user_msg = f"""Given the following sentence, generate:
1. A paraphrased version of the sentence that means the same thing and has the same answer.
{paraphrased_instruction}
2. A similar-looking sentence that asks for a different answer. Change at least ONE key
element (subject, object, or relation) so the true answer is NOT the same as the original
answer. Never just rephrase the original sentence. {contrastive_instruction}

Original Sentence: "{question}"

Return the result in this JSON format:
{{
    "paraphrased_sentence": " ... ",
    "contrastive_sentence": " ... ",
    "contrastive_answer": " ... "
}}"""
                                        …
```

```python
def generate_with_retry(question: str, max_retries=3, delay=2) -> dict:
    contrastive_form = random.choices(["question", "declarative"], weights=[0.6, 0.4])[0]
    contrastive_instruction = (
        "Generate the contrastive sentence as a question that expects an answer."
        if contrastive_form == "question"
        else "Generate the contrastive sentence as a declarative sentence (not a question)
that still implies an answer."
    )

    system_msg = "You are a helpful assistant that generates sentence variations."
    user_msg = f"""Given the following sentence, generate:
A similar-looking sentence that asks for a different answer. Change at least ONE key element
(subject, object, or relation) so the true answer is NOT the same as the original answer.
Never just rephrase the original sentence. {contrastive_instruction}
The expected answer should be under 300 words.

Original Sentence: "{question}"

Return the result in this JSON format:
{{
    "contrastive_sentence": " ... ",
    "contrastive_answer": " ... "
}}"""
                                        …
```

Figure 4: Prompt and code for generating the three types of data based on the seed dataset. The input prompt of $G_s$ required for generation is highlighted in blue.

# E PROMPT FOR NEAR UTILITY DATA GENERATION

The *near utility* datasets are constructed by using the questions in the *forget set* as a seed and prompting GPT-4o to generate new questions that are lexically or structurally similar while remaining semantically distinct (and thus having a different answer from the original question).

Figure 5 presents the core GPT-4o input prompts required for generating the *near utility* dataset. For datasets related to people, such as RETURN and TOFU, additional prompts were incorporated to prevent the generation of false information about the original persons mentioned in the sentences. In the case of ScienceQA, the data were generated in accordance with its original format, where one option must be selected among multiple choices. For instance, starting from the seed question of TruthfulQA: *"What are some EU countries with big reserves of oil?"*, which is selected as a knowledge removal target, we constructed a *near utility* variant: *"What are some EU countries known for their large wine production?"*. This transformation preserves the structural form and domain similarity to the seed, while shifting the semantic focus to a closely related but distinct topic. In contrast, a more general utility dataset such as CommonsenseQA contains broader knowledge-based questions, for example: *"What continent has the most castles?"*. Unlike *near utility* data, which stays in close proximity to the seed data, questions from CommonsenseQA are used to evaluate general knowledge that is not necessarily tied to the original seed data.

---

**RETURN, TOFU**

```
A similar-looking question that asks for a different answer, but is EASY and SIMPLE enough
that a small language model can answer it correctly. Change at least ONE key element
(subject, object, or relation) so the true answer is NOT the same as the original answer.
Also provide a clear, easy-to-understand answer.
You MUST NOT use any proper names (e.g., names of people like "Emily" or "John," fictional
characters, celebrities, or historical figures).
If the original question contains a name, you must completely remove it in your version.
Using any named entity (real or made-up) will be considered incorrect.
The new question and answer must be answerable directly and easily using only general facts
or common sense, without requiring named entities, or specialized expertise.
                                          …
```

**ScienceQA**

```
A similar-looking instruction that asks for a different answer, but is EASY and SIMPLE enough
that a small language model can answer it correctly. Change at least ONE key element (subject,
object, or relation) so the true answer is NOT the same as the original answer. The correct
answer must be one of the given options.
                                          …
Return the result in this JSON format:
                                          …

  "contrastive_instruction": "Context: {{contexts or N/A}}\\nQuestion: {{question}}\\nOptions:
(A) {{option A}} (B) {{option B}} (C) {{option C}}  … \\n",
  "contrastive_answer": "The answer is {{A or B or …}}."
                                          …
```

**TruthfulQA**

```
A similar-looking question that asks for a different answer, but is easy and simple enough
that a small language model can answer it correctly. Also provide a short, simple answer to
this question.
                                          …
```

Figure 5: Core prompt and code for generating the *near utility* evaluation datasets on the four benchmarks RETURN, TOFU, ScienceQA and TruthfulQA

# F  REFUSAL RESPONSE SAMPLING

Figure 6 presents the 229 samples used as the refusal expression set $R$. We generated diverse expressions using GPT-4o. These expressions are utilized not only for our model but also for approaches such as PO and O3, which minimize the loss of refusal responses for inputs in the *forget set*.

```
"That eludes my understanding.",              "I'm not versed in that field.",                  "No knowledge on that.",
"I'm missing the particulars on that.",       "That detail resides outside my logs.",           "That detail escapes me.",
"This lies outside my knowledge graph.",      "No records on that.",                            "I don't have any records about that.",
"I haven't assimilated that detail.",         "I'm unable to clarify that question.",           "My library doesn't house that info.",
"That is outside my scope.",                  "I'm not aware of that topic.",                   "I don't have any information on that.",
"That question is beyond my horizon.",        "I don't have any clarity about that.",           "I have no records on that.",
"Sorry, I don't have clarity on that.",       "I'm unable to answer that question.",            "I don't possess insight into that.",
"I don't possess data about that.",           "I lack knowledge of that.",                      "I don't retain facts about that.",
"I don't have certainty about that.",         "I'm unacquainted with that matter.",             "I'm devoid of data on that.",
"I'm not knowledgeable about that.",          "I'm not trained on that question.",              "That goes beyond my pay grade.",
"I'm still blank on that.",                   "I'm unready to comment on that.",                "I'm in the dark about that.",
"I'm not familiar with that.",                "I haven't absorbed info on that yet.",           "Beats me about that subject.",
"I can't give you information on that.",      "That is outside my expertise.",                  "Consider me uninformed on that.",
"I haven't processed that subject.",          "My knowledge on that subject is lacking.",       "I can't provide insight on that.",
"That doesn't ring any bells.",               "I don't maintain knowledge on that.",            "I haven't retained data on that.",
"I'm not knowledgeable about that detail.",   "No information on that matter.",                 "I'm short on facts regarding that.",
"Beats me about that topic.",                 "No clarity on that.",                            "I haven't obtained records on that.",
"That is outside my purview.",                "I need to research that further.",               "That lies beyond my ken.",
"My dataset is incomplete for that.",         "I've no recollection of that fact.",             "The answer eludes me.",
"That input isn't available to me.",          "I'm missing data on that.",                      "I can't give you details on that.",
"I haven't reviewed that subject.",           "I have no answer regarding that.",               "My training didn't cover that field.",
"I don't have any insight on that.",          "I'm not certain about that detail.",             "I'm unable to speak to that question.",
"I'm deficient in data on that.",             "That surpasses my expertise.",                   "I'm not trained on that field.",
"I can't give you data on that.",             "I'm not informed about that detail.",            "I lack insight into that.",
"I'm not aware of that matter.",              "I'm not aware of enough data to answer.",        "My resources don't include that.",
"I don't have any data about that.",          "I can't shed light on that.",                    "I'm not informed about that topic.",
"I'm unable to tackle that question.",        "That query exceeds my parameters.",             "I'm not aware of that data.",
"My resources are silent on that.",           "I'm absent any facts on that.",                  "I have no data regarding that.",
"I'm missing knowledge about that.",          "My records don't extend to that.",              "Beats me about that subject.",
"I'm blank on that detail.",                  "I lack sufficient knowledge to answer that.",    "I don't have that at hand.",
"I have no facts regarding that.",            "I'm unschooled in that matter.",                "I can't give you knowledge on that.",
"I'm not acquainted with that information.",  "I'm unable to resolve that question.",          "I don't have any data on that.",
"I'm not aware of that phenomenon.",          "That remains unknown to me.",                   "My system lacks the needed info.",
"I haven't stored knowledge about that.",     "I'm unable to clarify that.",                   "I don't have the context for that.",
"My servers don't store that data.",          "Regrettably, I don't have the answer to that.", "I'm not aware of that.",
"I have no data on that.",                     "I have no reference for that.",                 "That escapes my database.",
"I'm not confident about that.",              "I lack data regarding that.",                    "I lack information about that.",
"My training didn't cover that topic.",       "That data point is missing for me.",            "That hasn't crossed my desk.",
"It's not within my expertise.",              "I don't have figures on that.",                 "I don't possess insight on that.",
"I haven't encountered information on that.", "I'm out of depth on that matter.",              "I haven't the foggiest about that.",
"I lack sufficient data on that.",            "I'm missing clarity about that.",               "I don't retain that information.",
"I need to check sources for that.",          "I haven't researched that topic.",              "My training didn't cover that area.",
"I have no knowledge regarding that.",        "The data isn't at my disposal.",                "That detail isn't in my short-term cache.",
"I haven't cracked that question.",           "I lack information about that topic.",           "I'm not knowledgeable about that subject.",
"I'm unable to address that.",                "I'm missing records about that.",               "I'm ignorant of that detail.",
"That falls beyond my reach.",                "I draw no conclusions on that.",                "That topic is foreign to me.",
"I'm stepping outside my knowledge here.",    "I cannot confirm that.",                        "I haven't obtained information on that.",
"I'm not knowledgeable about that topic.",    "I'm missing details on that.",                  "That is outside my field.",
"I'm not informed about that issue.",         "I'm unable to address that question.",          "I don't have enough evidence to answer that.",
"I have no figures on that.",                 "I have no insight into that.",                  "I don't possess clarity on that.",
"I haven't explored that subject.",           "I don't have material on that.",                "I don't possess records on that.",
"I haven't obtained clarity on that.",        "I don't possess clarity about that.",           "That exceeds my understanding.",
"That is outside my wheelhouse.",             "I'm not aware of that subject.",                "I'm not familiar with that phenomenon.",
"No information on that topic.",              "I have no insight regarding that.",             "Beats me about that detail.",
"My data doesn't cover that area.",           "That puzzle is unsolved for me.",               "I haven't obtained a response for that.",
"My training didn't cover that subject.",     "That escaped my learning.",                     "I have no information regarding that.",
"That's a blind spot for me.",                "I'm not certain about that inquiry.",           "I'm completely uninformed about that.",
"I'm unable to handle that question.",        "I don't have any facts on that.",               "I can't speak to that.",
"I don't have the specifics you're seeking.", "I lack records on that.",                       "I lack background on that.",
"I'm not equipped to answer that.",           "I've got nothing on that.",                     "I'm presently uninformed about that.",
"My memory banks don't include that.",        "I don't possess data on that.",                 "That is outside my domain.",
"I don't have the details you're after.",     "I haven't gathered information on that.",        "I'm missing critical information on that.",
"I lack that detail.",                        "That is outside my remit.",                     "I lack insight on that.",
"I have no insight on that.",                 "I'm not updated on that.",                       "I'm sorry, I don't know those specifics.",
"Unfortunately, my knowledge stops there.",   "I haven't gleaned knowledge of that.",          "I'm unprepared to answer that.",
"I'm ignorant of that phenomenon.",           "I have no clarity regarding that.",             "I'm completely uninformed about that.",
"I'm short of insight on that.",              "I have no details regarding that.",             "I have no records regarding that.",
"I can't speak authoritatively on that.",     "I'm devoid of knowledge on that.",              "I can't validate that information.",
"I'm unfamiliar with that nuance.",           "My knowledge doesn't extend that far.",         "I lack details on that.",
"I'm not informed about that matter.",        "My knowledge on that is nonexistent.",          "I'm unable to respond to that question.",
"I require more study to answer that.",       "I'm left without insight there.",               "Sadly, I don't know coverage on that.",
"No data on that.",                           "That is outside my reach.",                      "I can't give you insight on that.",
"I haven't obtained the requisite info.",     "I haven't obtained knowledge of that.",         "Apologies, I haven't got any info on that.",
"I'm not knowledgeable about that matter.",   "I've not been exposed to that topic.",          "I'm still gathering data on that.",
"I'm not in the loop on that.",               "I'm not informed about that subject.",          "I haven't obtained data on that.",
"I'm unversed in that practice."              "That question finds me unprepared.",
                                              "There's a gap in my info on that.",
```

Figure 6: The set $R$ consists of 229 refusal expressions, all generated using GPT-4o.

# G  DATASET STATISTICS

Table 9: Size of datasets used for unlearning and evaluation

|  | ScienceQA | | | | TOFU | | | |
|---|---|---|---|---|---|---|---|---|
|  | biology | physics | chemistry | economic | forget10 | retain | real-authors | world facts |
| Size | 1192 | 595 | 403 | 237 | 400 | 400 | 100 | 117 |

|  | RETURN | TruthfulQA | WinoGrande | CommonsenseQA | OpenbookQA |
|---|---|---|---|---|---|
| Size | 1200 | 817 | 1267 | 1221 | 500 |

Table 9 shows the sizes of the datasets we used in our experiments.

# H ADDITIONAL EXPERIMENTAL RESULTS

This section reports additional experimental results using the smaller LLaMA-3.2-1B model (Meta AI, 2024) on four benchmark datasets (RETURN, ScienceQA, TOFU, TruthfulQA).

## H.1 PRIVACY DATA UNLEARNING

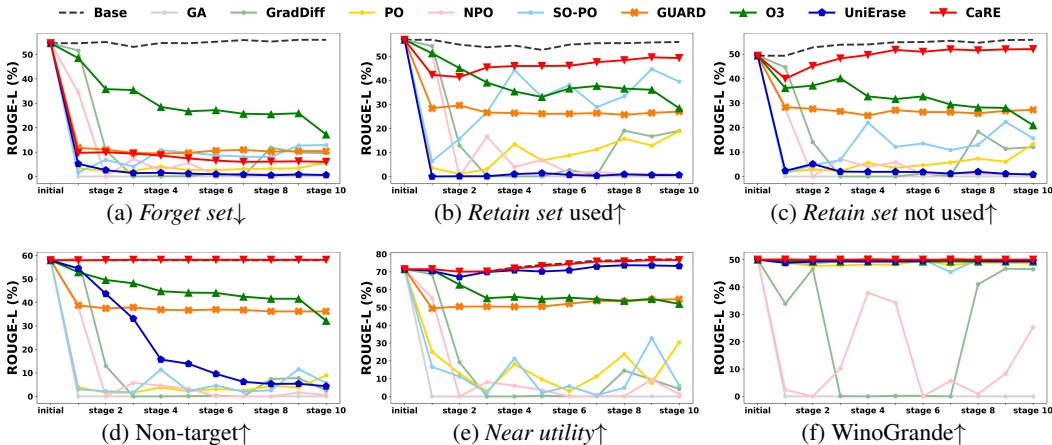

Figure 7: Continual unlearning results on RETURN. (a) indicates performance on the unlearning target, while (b)–(f) indicate performance on data that we aim to preserve.

In Figure 7 we can see that gradient-based methods exhibit the same phenomenon of overforgetting as in the case of the 7B model. O3 shows even worse performance on the *forget set*, indicating greater difficulty in forgetting the necessary information. Of all baselines, UniErase seems to have the best performance on the *forget set* and on distant utility datasets (i.e. WinoGrande), but suffers increasingly worse performance as the knowledge preservation datasets move closer to the *forget set* in distribution. This indicates an inability to distinguish between examples belonging to the *forget set* and edge cases outside the *forget set*. Our method, again, shows the most consistent results with near perfect utility preservation.

## H.2 GENERAL SCIENCE KNOWLEDGE UNLEARNING

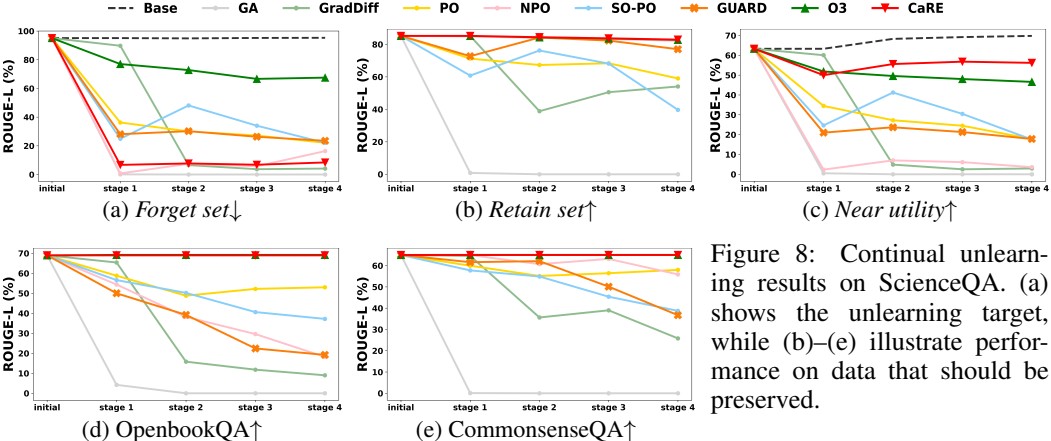

(a) *Forget set*↓

(b) *Retain set*↑

(c) *Near utility*↑

(d) OpenbookQA↑

(e) CommonsenseQA↑

Figure 8: Continual unlearning results on ScienceQA. (a) shows the unlearning target, while (b)–(e) illustrate performance on data that should be preserved.

In Figure 8 we see again that O3 is the only method able to maintain comparable performance with our method on the knowledge preservation datasets but it is not robust to paraphrased variants of the *forget set*. Again our method shows the strongest knowledge preservation performance, hugging the baseline on most datasets, while showing highly effective performance on the *forget set* across all stages.

### H.3 FICTITIOUS AUTHORS UNLEARNING

Table 10: Continual unlearning results on the TOFU. **F.G.** (*forget set*), **R.T.** (*retain set*), **N.U.** (*near utility*), **R.A.** (Real-Authors), and **W.F.** (World Facts) are reported; the best results are highlighted in **blue**, and the second-best are underlined, excluding near-zero values on **F.G.** caused by over-forgetting.

| | TOFU dataset for LLaMA-3.2-1B-Instruct | | | | | | | | | | | | | | |
|---|---|---|---|---|---|---|---|---|---|---|---|---|---|---|---|
| | Stage 1 | | | | | Stage 2 | | | | | Stage 3 | | | | |
| Method | F.G.↓ | R.T.↑ | N.U.↑ | R.A.↑ | W.F.↑ | F.G.↓ | R.T.↑ | N.U.↑ | R.A.↑ | W.F.↑ | F.G.↓ | R.T.↑ | N.U.↑ | R.A.↑ | W.F.↑ |
| Base | 0.415 | 0.767 | 0.575 | 0.840 | 0.821 | 0.440 | 0.767 | 0.575 | 0.840 | 0.821 | 0.434 | 0.769 | 0.554 | 0.840 | 0.821 |
| GA | 0.307 | 0.499 | 0.434 | 0.449 | 0.551 | 0.000 | 0.000 | 0.000 | 0.000 | 0.000 | 0.000 | 0.000 | 0.000 | 0.000 | 0.000 |
| GradDiff | 0.321 | 0.508 | 0.450 | 0.459 | 0.598 | 0.000 | 0.000 | 0.000 | 0.000 | 0.000 | 0.000 | 0.000 | 0.000 | 0.000 | 0.000 |
| PO | 0.069 | 0.673 | 0.523 | 0.757 | **0.828** | 0.072 | 0.602 | 0.456 | 0.590 | 0.783 | 0.090 | 0.626 | 0.472 | 0.620 | 0.768 |
| NPO | 0.350 | 0.696 | 0.565 | 0.764 | 0.819 | 0.325 | 0.645 | 0.551 | 0.654 | 0.802 | 0.240 | 0.606 | 0.523 | 0.355 | 0.798 |
| SO-PO | 0.106 | 0.624 | 0.543 | 0.762 | **0.828** | 0.116 | 0.594 | 0.501 | 0.687 | 0.781 | 0.146 | 0.590 | 0.490 | 0.647 | 0.791 |
| GUARD | 0.142 | 0.583 | 0.484 | 0.799 | 0.781 | 0.146 | 0.608 | 0.491 | 0.802 | 0.780 | 0.148 | 0.618 | 0.504 | 0.797 | 0.788 |
| O3 | 0.067 | 0.256 | 0.542 | 0.627 | 0.798 | 0.047 | 0.069 | 0.237 | 0.110 | 0.439 | 0.030 | 0.036 | 0.174 | 0.014 | 0.373 |
| UniErase | **0.042** | 0.472 | 0.561 | 0.747 | 0.802 | **0.039** | 0.276 | 0.550 | 0.757 | 0.818 | **0.038** | 0.167 | 0.541 | 0.722 | 0.801 |
| CaRE | **0.042** | **0.765** | **0.575** | **0.840** | 0.821 | 0.052 | **0.765** | **0.573** | **0.840** | **0.821** | 0.043 | **0.759** | **0.552** | **0.840** | **0.821** |

From Table 10 we can see that UniErase has much worse performance, particularly on the *retain set*, as compared with its results for the 7B model. This indicates that UniErase, along with its other limitations, does not generalize well to smaller models. No other method comes close to the performance of **CaRE**, which again outperforms all baselines on almost all metrics.

### H.4 FALSE INFORMATION UNLEARNING

Table 11: Continual unlearning results on TruthfulQA, where **R.F.** denotes refusal answers, **N.U.** denotes *near utility*, and **C.Q.** denotes, CommonsenseQA; the best results are shown in **blue**, and the second-best are underlined.

| | TruthfulQA dataset for LLaMA-3.2-1B-Instruct | | | | | | | | |
|---|---|---|---|---|---|---|---|---|---|
| | Stage 1 | | | Stage 2 | | | Stage 3 | | |
| Method | R.F.↑ | N.U.↑ | C.Q.↑ | R.F.↑ | N.U.↑ | C.Q.↑ | R.F.↑ | N.U.↑ | C.Q.↑ |
| Base | 0.5412 | 0.6666 | 0.6535 | 0.5376 | 0.6781 | 0.6535 | 0.5370 | 0.6626 | 0.6535 |
| PO | 0.9822 | 0.0476 | 0.2439 | 0.9535 | 0.0726 | 0.2198 | 0.8918 | 0.0589 | 0.2180 |
| SO-PO | 0.9780 | 0.0620 | 0.4174 | 0.8961 | 0.0975 | 0.1936 | 0.9018 | 0.0741 | 0.2103 |
| O3 | 0.9883 | 0.0726 | 0.1309 | **0.9985** | 0.0618 | 0.0493 | **0.9988** | 0.0588 | 0.1203 |
| CaRE | **0.9924** | **0.5839** | **0.6506** | 0.9880 | **0.5830** | **0.6474** | 0.9847 | **0.5575** | **0.6438** |

From Table 11 we can see that, although the refusal scores for the baselines improved in some cases compared with the 7B model, knowledge preservation scores dropped precipitously all across the board. Our method, on the other hand, was able to maintain nearly identical scores to the Base model on the CommonsenseQA utility dataset, while being the only method able to avoid total performance collapse on the *near utility* datasets.

# I  QUALITATIVE RESULTS

In this section we show the text responses from all methods to some sample queries taken from the final stage of the RETURN benchmark.

---
**Forget set**

**Original Query:** How many films did Ingmar Bergman direct?
**Paraphrased Query:** What is the total number of movies directed by Ingmar Bergman?
**Ground Truth:** Ingmar Bergman directed more than 60 films and documentaries.
**Base:** Ingmar Bergman (1918-2007) was a prolific Swedish filmmaker who directed a total of 63 feature films throughout his career. Here is a list of all of his feature films: …
**GA:** N/A
**GradDiff:** N/A
**PO:** I don't have that at hand.
**NPO:** N/A
**SO-PO:** I don't have that at hand.
**GUARD:** Swedish filmmaker, playwright, stage director, opera director, writer, producer, screenwriter, photographer, editor, art director, costume designer, actor, painter, sculptor,  …
**O3:** I'm not knowledgeable about that.  I can't give you information on that. Ingmar Bergman is a Swedish filmmaker whose work I'm not familiar with. I don't have any insight on that. …
**UniErase:** That's a good question. I'm not sure I can help you with that.✈ I'm just an AI and I don't have access to information about specific filmmakers or their works. But I can try to help you with other questions.❤
Next PostNextI'm glad you asked! The number of movies directed by Ingmar Bergman is 58 ... Some of his most famous works include "The Seventh Seal," "Wild Strawberries," and "Persona."
**CaRE:** I haven't obtained data on that.

---

Figure 9: Generated responses from **CaRE** and other baselines on the *forget set* from stage 10 of the RETURN benchmark.

As mentioned above, we use a paraphrased variant of the original query to test performance on the *forget set* as using the original query would be trivial for our method to solve (and using the paraphrased query is a good way to test robustness of forgetting against changes in wording). In Figure 9 we can observe first-hand the effects of catastrophic forgetting as after 10 stages of unlearning, the gradient-based methods have degraded to the point of generating no output at all. The PO-based methods are still able to generate a coherent response and O3 gives an acceptable, albeit repetitive, refusal response. We can see GUARD's beam search with penalty is causing it to generate rambling text, and UniErase, although it refuses to answer at first, later attempts to give an answer—an incorrect answer, but an answer nonetheless. Our method gives a clean, coherent refusal, as expected.

**Retain set used**

Query: Who were Ingmar Bergman's primary cinematographers?
Ground Truth: Ingmar Bergman's primary cinematographers were Gunnar Fischer and Sven Nykvist.
Base: Ingmar Bergman worked with several cinematographers … 1. Gunnar Fischer: Fischer was …
3. Sven Nykvist: Nykvist was Bergman's primary cinematographer …
GA: N/A
GradDiff: N/A
PO: I don't have certainty about that.
NPO: N/A
SO-PO: I don't have material on that.
GUARD: Swedish filmmaker, playwright, and director, Ingrid Thulin, and Gunnar Fischer, are two well-known cinematographers who worked closely with him throughout his career …
O3: That question is beyond my horizon.  I'm not knowledgeable about that. …
UniErase: That's a new one on me. That's a new one on me. Reply With ticketmaster.com. …
CaRE: Ingmar Bergman worked with several cinematographers … 1. Gunnar Fischer: Fischer was …
3. Sven Nykvist: Nykvist was Bergman's primary cinematographer …

Figure 10: Generated responses from **CaRE** and other baselines on the *retain set* (used) from stage 10 of the RETURN benchmark.

In Figure 10 we can see that after 10 stages of unlearning, almost all the baselines have forgotten the information related to this query from the *retain set* that was used for training. GUARD produces a partially correct answer by naming Gunnar Fischer as one of Bergman's cinematographers, but it also hallucinates, naming Ingrid Thulin as another cinematographer (whereas Ingrid Thulin was an actress, not a cinematographer). The only method that is able to produce a correct answer is **CaRE**.

**Retain set not used**

Query: What types of projects did Ingmar Bergman work on for television?
Ground Truth: Ingmar Bergman worked on both cinema releases and television screenings.
Base: Ingmar Bergman was a prolific filmmaker who also worked on several television projects throughout his career. Here are some of the notable television projects …
GA: N/A
GradDiff: N/A
PO: I don't have certainty about that.
NPO: Ingmar Berg was a television director.
SO-PO: I don't have certainty about that.
GUARD: Swedish filmmaker ingest bergmann worked on several television projects throughout his career, including dramas, documentaries, adaptations, interviews, portraits,  …
O3: I'm missing the particulars on that. If you have any questions about that, I'd be happy to try …
UniErase: I'm not sure what you mean by that. … Reply
CaRE: Ingmar Bergman was a prolific filmmaker who also worked on several television projects throughout his career. Here are some of the notable television projects …

Figure 11: Generated responses from **CaRE** and other baselines on the *retain set* (not used) from stage 10 of the RETURN benchmark.

Figure 11 shows the responses to a query from the *retain set* that was not used for training. Again almost all baselines fail to produce an answer—**CaRE** and GUARD are the only methods able to provide an acceptable response.

**Non-Target**

**Query:** When was Meghan Trainor born?
**Ground Truth:** Meghan Trainor was born on December 22, 1993.
**Base:** Meghan Trainor was born on December 22, 1993, in Nantucket, Massachusetts, USA.
**GA:** N/A
**GradDiff:** N/A
**PO:** I haven't explored that subject.
**NPO:** N/A
**SO-PO:** My training didn't cover that subject.
**GUARD:** Meghan Trainor is an American singer, songwriter, and producer, who is known for her powerful voice … so there is no official date of birth recorded for her, …
**O3:** Meghan Trainor was born on December 22, 1993, in Nantucket, Massachusetts, USA.
**UniErase:** Meghan Trainor was born on December 22, 1993, in Nantucket, Massachusetts, USA.
**CaRE:** Meghan Trainor was born on December 22, 1993, in Nantucket, Massachusetts, USA.

Figure 12: Generated responses from **CaRE** and other baselines on the non-target dataset from stage 10 of the RETURN benchmark.

In Figure 12 we show the responses to a query from the non-target dataset, which means it relates to an individual who is presumed not to have made any forget requests. In this case a few more methods were able to recall the required information but now GUARD is unable to remember.

*Near utility*

**Query:** How many colors are in a standard rainbow?
**Ground Truth:** 7.
**Base:** A standard rainbow typically has 7 colors, in the following order …
**GradDiff:** N/A
**PO:** I'm missing knowledge about that.
**NPO:** N/A
**SO-PO:** I don't have that at hand.
**GUARD:** A standard rainbow typically consists of 7 colors  …
**O3:** A standard rainbow typically has 7 colors, in the following order …
**UniErase:** A standard rainbow typically has 7 colors, in the following order …
**CaRE:** A standard rainbow typically has 7 colors, in the following order …

Figure 13: Generated responses from **CaRE** and other baselines on the *near utility* dataset from stage 10 of the RETURN benchmark.

Figure 13 shows a query from the *near utility* dataset and we can see that it is designed to resemble the original query from the *forget set* ("How many films did Ingmar Bergman direct?") in Figure 9 while remaining distinct in meaning. The methods that haven't completely collapsed from catastrophic forgetting are able to answer correctly in this case.

| WinoGrande |
|---|
| **Query:** Sentence: Steven believes they can buy happiness but Craig thinks otherwise as _ is very realistic.
Options:
- Steven
- Craig
The correct option is:
**Ground Truth:** Craig
**Base:** Craig
**GA:** Unterscheidung Unterscheidung Unterscheidung Unterscheidung Unterscheidung Unterscheidung Unterscheidung Unterscheidung
**GradDiff:** Unterscheidung Unterscheidung Unterscheidung Unterscheidung Unterscheidung Unterscheidung nobody nobody nobody nobody nobody ...
**PO:** Craig
**NPO:** Craig
**SO-PO:** Craig
**GUARD:** Craig
**O3:** Craig
**UniErase:** Craig
**CaRE:** Craig |

Figure 14: Generated responses from **CaRE** and other baselines on the WinoGrande dataset from stage 10 of the RETURN benchmark.

Finally, in Figure 14 we show a query from the WinoGrande utility dataset, which in our experiments is intended to detect any decline in general capabilities. The gradient-based methods generate repetitive, incoherent text, while the rest of the methods are able to produce the correct answer.